# Formulation and Characterization of Edible Films Based on Organic Mucilage from Mexican *Opuntia ficus-indica*

**Dulce C. González Sandoval**, **Brenda Luna Sosa**,
**Guillermo Cristian Guadalupe Martínez-Ávila**, **Humberto Rodríguez Fuentes**,
**Victor H. Avendaño Abarca and Romeo Rojas \***

Research Center and Development for Food Industries, Campus of Agricultural Sciences, School of Agronomy, Universidad Autónoma de Nuevo León, Francisco Villa s/n, Ex−Hacienda "El Canadá", General Escobedo, Nuevo León 66050, Mexico
\* Correspondence: romeo.rojasmln@uanl.edu.mx; Tel.: +52-81-1340-4399 (ext. 3522)

**Abstract:** The consumption of organic products has increased in recent years. One of the most important products in Mexico is nopal. Nopal's content and properties make the formulation of edible films possible. In this study, we aimed to develop and characterize biodegradable edible films containing mucilage from *Opuntia ficus-indica*. The mucilage extraction yield, thickness, color, water vapor permeability, light transmission rate, film transparency, solubility, stability of dispersion, and puncture strength were measured. The use of mucilage from different cultivars affected the water vapor permeability ($8.40 \times 10^{-11}$ g·m$^{-1}$·s$^{-1}$·Pa$^{-1}$ for cultivar Villanueva, $3.48 \times 10^{-11}$ g·m$^{-1}$·s$^{-1}$·Pa$^{-1}$ for Jalpa, and $1.63 \times 10^{-11}$ g·m$^{-1}$·s$^{-1}$·Pa$^{-1}$ for Copena F1). Jalpa provided the most soluble mucilage with the highest thickness (0.105 mm). Copena F1 provided the clearest film with the greatest transparency (3.81), the best yellowness index, and the highest resistance (4.44 N·mm$^{-1}$). Furthermore, this film had the best light transmission rate (48.93%). The Copena F1 showed the best film formation solution viscosity. These results indicate that mucilage mixed with pectin is a potential source for the formulation of edible films.

**Keywords:** organic; mucilage; edible films; characterization

---

## 1. Introduction

In the last several years, interest in the production of edible films has increased because they present a series of advantages, such as prolonging the shelf life of foods and reducing the use of plastics. Therefore, many film-forming materials such as starch [1–4], gelatin [5], chitosan [6–8], cellulose [9], carboxymethyl cellulose [10], methylcellulose [11], pectin [12,13], agar [14], alginate [15,16], carrageenan [16], zein [17], etc., have been used. Mucilage from Mexican *Opuntia ficus-indica* is suitable for large-scale production due to its abundance in Mexico, low cost, nontoxicity, biodegradability, biocompatibility, film-forming capacity, and renewability. Several studies have evaluated the properties of films and coatings based on mucilage [18], mainly in combination with other materials to make film formation possible [19].

Nopal is a cactus, grown and harvested in arid and semiarid regions worldwide, with diverse industrial applications. It is widely cultivated and consumed in Mexico and was one of the first foods produced around the world for human consumption. Nopal has been traditionally used for preparing diverse local foodstuffs, powder, pickled or in brine, juice, and syrup [20]. In nopal processing, the mucilage is separated and removed from products because the sap may confer undesirable textural

and sensorial properties to the final product. Mucilage from the genus *Opuntia ficus-indica* is composed of heteropolysaccharide hydrocolloids (arabinose, galactose, galacturonic acid, rhamnose, xylose and pectin [21,22]) with a wide range of physicochemical properties, and calcium carbonate is present in two different crystalline forms. The importance of this is that the calcium in these compounds is bioavailable for the human body. The color is a yellow-green hue, the pH is slightly acidic, and non-Newtonian shear thinning behavior is present [23].

At present, the population has returned to the consumption of organic products. Therefore, there is a need to formulate edible coatings from organic sources. In nopal, the mucilage is and will remain an undesirable material, and it is therefore a potential source for use in the manufacture of edible films. However, the first stage is to formulate and characterize films with the help of other polymers such as pectin (in this case), to help with the polymer matrix formation by adding glycerol as a plasticizer. The characterization consists of evaluating the film's diverse properties, such as solubility, thickness, viscosity of the film-forming solution, water vapor permeability (WVP), thermal and mechanical properties, fourier transform infrared spectroscopy (FTIR) analysis, color, etc. [1,17,24,25]. The importance of this characterization is to understand how the formulation parameters (pH, concentration, additives, varieties, sources, etc.) affect the properties of the film in terms of its barrier, mechanical, physical, thermal, structural properties, etc., to demonstrate its high potential for use as a biodegradable packaging material in the food industry.

Recently, several authors have evaluated the use of mucilage from *Opuntia ficus-indica* for the formulation and characterization of edible films. Guadarrama-Ledezma et al. [20] formulated edible films with citric pectin and mucilage and found that as the mucilage concentration in films increased, the vapor water permeability decreased, due to the enhance internal cohesiveness of the components. Furthermore, the combination of mucilage and pectin led to enhanced thermal stability and higher water vapor permeability. Dominguez-Martinez et al. [22] developed edible films containing mucilage, chitosan, and polyvinyl alcohol in different concentrations. The results suggested that the interaction between chitosan and mucilage can increase water vapor permeability. Espino-Díaz et al. [26] extracted mucilage from *Opuntia ficus-indica* and prepared film formation dispersions with different pH values and calcium concentrations. The authors found that the addition of calcium increased the WVP, and that the pH affected the mechanical properties of the film. Rodriguez et al. [19] evaluated the influence of edible coatings based on linseed mucilage, alginate, and fructooligosaccharide containing *Lactobacillus casei LC*-01 on the shelf life of fresh-cut yacon cubes. They found that the use of these films helped preserve the physicochemical parameters of the vegetable, reducing weight loss and darkening. Morais et al. [27] evaluated edible coatings based on spineless cactus mucilage. The authors found that biocoating with mucilage reduced dehydration and maintained the visual and sensory qualities of yam slices.

The aim of our current study is to evaluate formulations of edible films based on organic mucilage from *Opuntia ficus-indica* and pectin. We evaluated the films' main properties in order to determine the best option for application to food products with the aim of preserving their quality and shelf life.

## 2. Materials and Methods

### 2.1. Plant Material

Three cultivars of *Opuntia ficus-indica*—Jalpa, Villanueva and Copena F1—12 months of age were used from the germplasm bank (School of Agronomy, Campus Marin, Universidad Autónoma de Nuevo León (UANL), Nuevo León, México).

### 2.2. Mucilage Extraction

The extraction was carried out in the chemical and biochemical lab (School of Agronomy, UANL) according to the method by Espino-Diaz et al. [26]. The cladodes were washed with a detergent solution (Axion® 1% *v/v*). The thorns were removed with a knife and weighed on an analytic balance

(TS-M Series Thomas®, Swedesboro, NJ, USA). The cladodes were then ground in a blender (Model 465015, Oster, New York, NY, USA), put into a glass vessel, and heated for 1 h at 80 °C to separate the mucilage from the fiber, and eliminate the enzymatic activity. The vessel was placed into an ice bath to reduce the temperature to 25 °C. The fiber was separated using mousseline cloth, and the mucilage was precipitated with absolute ethanol at a ratio of 2:1. The precipitate was filtered using a vacuum pump (Model AIT-2B, GB®, Iztapalapa, México) in two steps. The filtered product was placed into Petri dishes and dried in a stove for 24 h at 60 °C. We used a mortar to grind the dry sample into a fine powder. The mucilage was placed into a black container until use.

### 2.3. Formulation of Edible Films

For each of the cultivars, two aqueous solutions were prepared. The first contained mucilage powder and distilled water at a ratio of 1:100 mixed with constant stirring for 24 h at room temperature. The second contained pectin and distilled water at a ratio of 1.5:100 mixed with constant stirring for 40 min at room temperature. Both solutions were mixed at a ratio of 1:1 with 2 mL of glycerol as a plasticizer for 30 min with constant stirring. A total of 28 mL from each solution was transferred to a Petri dish with a diameter of 8.5 cm. The solutions were dried in a stove at 45 ± 2 °C for 18 h. The edible films were retrieved from the Petri dishes and stored in hermetic bags until characterization. Each edible film was prepared in triplicate.

### 2.4. Characterization of Edible Films

#### 2.4.1. Thickness

Thickness was measured using a hand-held micrometer (Model IP65 0-1″, Mitutoyo America Corporation, Aurora, IL, USA). Ten thickness measurements were taken for each triplicate film prepared.

#### 2.4.2. Color

Total differences in color *(ΔE)* and yellowness index *(YI)* were determined according to the procedure by Rhim et al. [25], with a colorimeter (Konica Minolta, Chroma Meter CR-400/410, Tokyo, Japan) and using the Commission Internationale de l'Eclairage CIELab scale. From the *L\**, *a\** and *b\** parameters, *ΔE* was calculated using Equation (1). The values of the white standard plate were *L\** = 96.9, *a\** = −0.04 and *b\** = 1.84. The *YI* was calculated according to Pérez-Mateos et al. [24] using Equation (2). Five measurements were taken from each treatment, with three determinations from different areas of the edible film.

$$\Delta E = \sqrt{\left(L^* - L^*_{\text{reference}}\right)^2 + \left(a^* - a^*_{\text{reference}}\right)^2 + \left(b^* - b^*_{\text{reference}}\right)^2} \tag{1}$$

$$YI = 142.86 \, \frac{b^*}{L^*} \tag{2}$$

#### 2.4.3. Water Vapor Permeability (WVP)

WVP ($g \cdot m^{-1} \cdot s^{-1} \cdot Pa^{-1}$) was calculated using the gravimetric method ASTM E96-9 and Equation (3) according to Zhang et al. [28] with few modifications. A test film was placed on an acrylic cup containing $CaCl_2$ and sealed. The whole assembly was weighed and placed in a chamber containing 500 mL of distilled water. The chamber was equipped with a fan and a digital relative humidity (RH)-meter. The relative humidity inside the chamber was kept at around 99% and the temperature at 20 °C during the experiment. The fan inside the chamber was turned on during the entire test period to ensure that the relative humidity inside the chamber was even. The weight gain of the cups over time was measured periodically (each hour for 6 h) to obtain the water vapor transmission rate (WVTR) of the films. In calculating the WVTR, linear regressions ($R^2 > 0.99$) were accomplished between weight gain and time using regression options in spreadsheet software. Three replications were conducted for

the same treatment. The means of these four thickness values were used to calculate the WVP from the WVTR results:

$$\text{WVP} = \frac{\text{WVTR}}{A} \frac{e}{\Delta Pv} \tag{3}$$

where WVP is the water vapor permeability, WVTR is the water vapor transmission rate ($\text{g·s}^{-1}$) obtained as the slope of the linear regression of time versus weight gain; $e$ is the mean of the thickness of the film (m); $A$ is the area of WVT ($\text{m}^2$); and $\Delta Pv$ is the difference in water vapor pressure between the atmosphere of $CaCl_2$ and that of the chamber (2337 Pa). More specifically, WVTR is the slope obtained by plotting the final weight minus the initial weight of the sample ($W_f - W_0$) versus time ($t$).

### 2.4.4. Light Transmission Rate and Film Transparency

Light transmission rate and film transparency were measured according to methods by Zhang et al. [28], Shiku et al. [29] and Fang et al. [30] The selected wavelengths were from 200 to 800 nm using a UV/visible spectrophotometer. Each edible film (0.5 cm × 4.0 cm) was placed in a quartz cell to measure the absorbance and transmittance. The transparency was calculated using Equation (4) according to Han et al. [31]:

$$\text{Transparency} = \frac{A_{600}}{s} = \frac{-\log T_{600}}{s} \tag{4}$$

where $A_{600}$ and $T_{600}$ are the absorbance and transmittance at 600 nm, respectively, and $s$ is the thickness of the film.

### 2.4.5. Solubility

According to Pinotti et al. [32], 2 cm × 3 cm pieces of each film were cut and stored in a desiccator with silica gel for 7 days. Samples were weighed and placed into test beakers with 80 mL of deionized water. The samples were maintained under constant agitation at 200 rpm for 1 h at 25 °C. The remaining pieces of film were then collected by filtration and dried again in an oven (at 60 °C for 24 h) to constant weight. The percentage of total soluble matter (% solubility) was calculated as follows:

$$\% \, \text{Solubility} = \frac{(\text{Initial dry weight} - \text{Final dry weight})}{\text{Initial dry weight}} \times 100 \tag{5}$$

Samples were analyzed in at least duplicate.

### 2.4.6. Stability of Dispersion

Five replicates per treatment were taken from each solution (15 mL each) and put into Falcon tubes. The samples were kept for 72 h at room temperature (25 °C) or under refrigeration (7 °C). Furthermore, an accelerated assay under centrifuge (Hermle Labortechnik GmbH Z400K, Gosheim, Germany) was performed at 3500 rpm/10 min to prove the presence or absence of cremation/phase separation.

### 2.4.7. Viscosity

From each solution, 200 mL were taken and placed into a rotational viscosimeter of concentric cylinders (Brookfield model DV-E, Engineering Laboratories, Inc., Oakland, NJ, USA). The spindle was selected by taking into account that the percentage of torque should not be less than 10% for each variety. The apparent viscosity was measured by varying the rotational speed in ascending order (20, 30, 50 and 60 rpm). All measurements were made at 26 °C with five replicates per treatment.

### 2.4.8. Puncture Strength (PS)

Five pieces of edible film, 3 cm in diameter, were cut. The edible film was placed on a 10 mL glass vessel and held with links at the top. The resistance was measured in $\text{N·mm}^{-1}$ by puncture with a

texture analyzer (TA.XT plus, Stable Micro Systems, Godalming, England). A cylindrical tool P/2 (2 mm diameter) descending at a distance of 10 mm with a speed of 2 mm·s$^{-1}$ was used.

### 2.4.9. Statistical Analysis

The data obtained were analyzed as a completely random design with InfoStat software (version 2017 1.2) using variance analysis. A comparison of means was made by Tukey's test ($p \leq 0.05$).

## 3. Results and Discussion

Water plays a key role in the deterioration of foods, and thus, an important characteristic of edible films is their ability to prevent the exchange of moisture between the surroundings and the food matrix [33]. WVP is the amount of moisture that passes through a unit area of material per unit time, and low values are related to an extended shelf life of the products (Figure 1) [34]. The edible films analyzed showed low WVP values with significant differences between films from different cultivars. The highest WVP was for the cultivar Villanueva ($8.40 \times 10^{-11}$ g·m$^{-1}$·s$^{-1}$·Pa$^{-1}$), followed by Jalpa ($3.48 \times 10^{-11}$ g·m$^{-1}$·s$^{-1}$·Pa$^{-1}$), then Copena F1 ($1.63 \times 10^{-11}$ g·m$^{-1}$·s$^{-1}$·Pa$^{-1}$). These values are similar to those reported for other biomaterials, such as a film of pectin-lemon essential oil-glycerol ($5.17 \times 10^{-11}$ g·m$^{-1}$·s$^{-1}$·Pa$^{-1}$) by Sánchez-Aldana et al. [35] and a film of garlic powder-cassava starch-glycerol ($7.5 \times 10^{-10}$ g·m$^{-1}$·s$^{-1}$·Pa$^{-1}$) by Famá et al. [36]. This property is one of the most important in terms of film characterization because it gives an idea of how much the film will help in counteracting water loss from the product to be evaluated. Films must have low permeability to reduce the product's moisture loss [37]. The products' deteriorative processes are intrinsically related to water, which is why the WVP of edible films must be taken into account [35]. The additives in the formulations also contribute fundamentally to the films' characteristics. One of these additives is glycerol, which is added as a plasticizer, but the amount added must be considered carefully because glycerol is very hygroscopic. This would facilitate the film's interaction with water and therefore increase its water absorption capacity, making it more permeable and thus having lower barrier properties [38]. Furthermore, these low values of WVP could be explained by better reorganization of the polymer chains resulting from longer drying times [1].

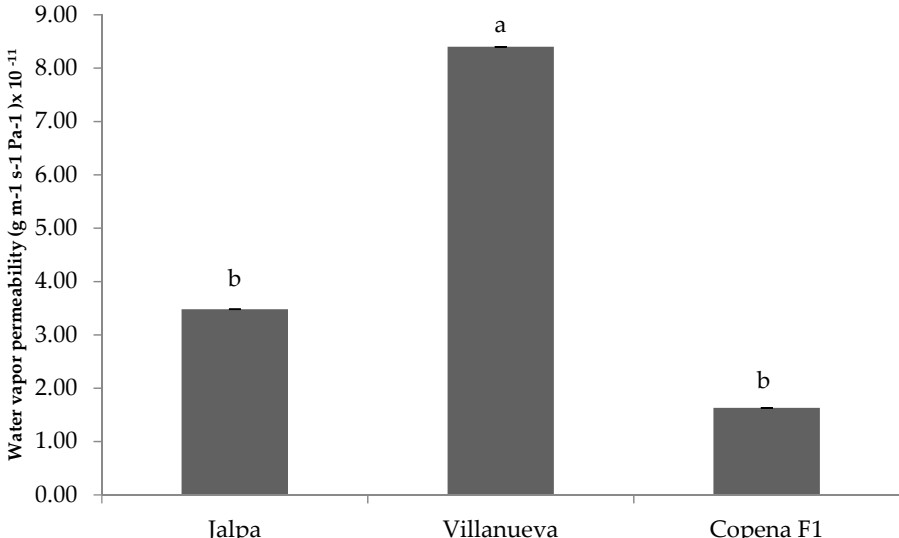

**Figure 1.** Water vapor permeability (WVP) of mucilage-based films made using three cultivars of *Opuntia ficus-indica*. Columns with the same letter are significantly ($p \leq 0.05$) different. The bars represent the standard error.

The other evaluated physicochemical properties of the films were thickness, solubility, transparency, stability of the dispersion, total color difference, and yellowness index (Table 1). This information provides indications regarding the films' behavior when they are applied to the product. The cultivar Jalpa film had the highest level of thickness (0.105 mm); Villanueva and Copena F1 had lower thickness values compared to Jalpa by 25.71% and 23.80%, respectively. These values differ from those reported by González [39] for a film based on *Opuntia ficus-indica* mucilage-glycerol-polyethylene glycol-oleic acid (0.222 mm). This is mainly due to the number of solutes incorporated into the film formulation [40].

**Table 1.** Physicochemical properties of the edible films based on the mucilage of three cultivars of *Opuntia ficus-indica*.

| Property | *Jalpa | *Villanueva | *Copena F1 |
|---|---|---|---|
| Thickness (mm) | 0.105 ± 0.003a | 0.078 ± 0.005b | 0.080 ± 0.005b |
| Solubility (%) | 91.036 ± 1.095a | 89.858 ± 4.955a | 81.266 ± 0.785b |
| Light transmission rate (%) | 18.48 ± 1.84b | 44.50 ± 0.42a | 48.93 ± 2.31a |
| Transparency | 7.431 ± 0.895a | 4.674 ± 0.364b | 3.817 ± 0.288b |
| ΔE | 18.684 ± 1.409a | 15.440 ± 0.496b | 11.826 ± 0.185c |
| YI | 24.999 ± 2.553a | 21.618 ± 0.716b | 14.704 ± 0.183c |
| SD | PS | PS | PS |

*Cultivar; SD, stability of dispersion; PS, phase separation; ΔE, total color difference; YI, yellowness index; data are presented as mean ± SD. Values in rows with different letters are significantly ($p \leq 0.05$) different.

The solubility of the materials is important because the coating will be consumed with the product, that is, it is an edible matrix [35]. There were significant differences ($p < 0.05$) in this property between the cultivars. The most soluble film was that of the cultivar Jalpa (91.03%), followed by Villanueva (89.85%) and Copena F1 (81.26%). Low solubility indicates that there is greater cohesion in the polymer matrix, mainly due to the formation of numerous hydrogen bonds between the chains of the polymers involved [41] due to the affinity of the components of the edible film or coating towards water. Hence, determining this parameter is useful for predicting potential applications [1].

The color and yellowness index are important parameters affecting the packaging facade, consumer acceptance, and food quality [10]. Significant differences were found in the total color and yellowness index among the cultivar films, with those of the cultivar Jalpa being the highest. According to these parameters, the films of the cultivar Copena F1 were clearer than those of the other two cultivars. These properties are related to the optical properties of the material, and therefore, must be taken into account at the time of its application and evaluation. According to the stability test, the three dispersions maintained their stability at the beginning but presented phase separation after the test in the centrifuge. This is possibly due to the low amount of glycerol that was used in the formulation. According to De Ancos et al. [42], the addition of plasticizers helps improve the mechanical and stabilizing properties of the formulations, improving their stability.

The mechanical properties determine the ability of the edible films to maintain the integrity of the packed product and the strength of the material [43]. The behavior of the films against puncture force is shown in Figure 2. The cultivar that produced the film with the greatest resistance to the force exerted was Copena F1 (4.44 N·mm$^{-1}$), followed by Jalpa (3.95 N·mm$^{-1}$) and Villanueva (3.83 N·mm$^{-1}$). These results indicate low values of resistance to puncture force, which is possibly due to the amounts of film-forming elements, mainly the amount of mucilage. Polysaccharides have low functional properties, but when mixed with proteins, these properties may increase [44]. Furthermore, an increase in the plasticizer concentration reduces intermolecular forces, thereby increasing flexibility and reducing the materials' strength [45–47].

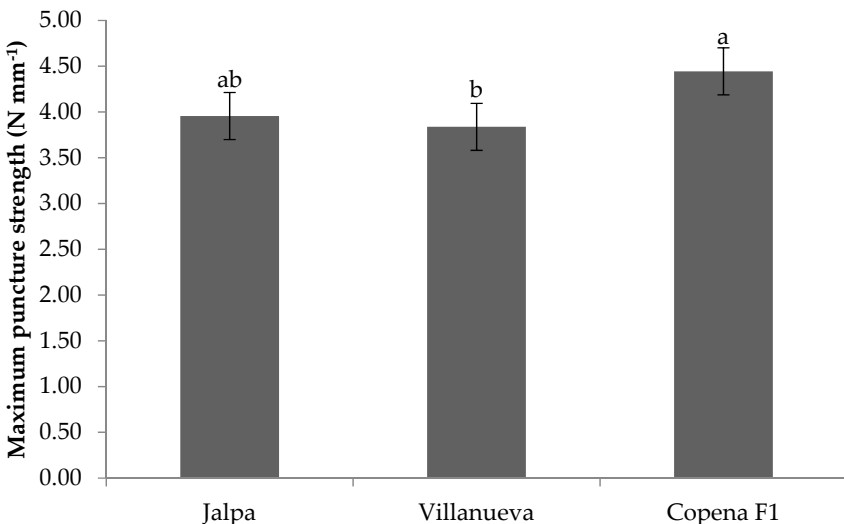

**Figure 2.** Maximum puncture strength (N·mm$^{-1}$) of the edible films based on the mucilage of three cultivars of *Opuntia ficus-indica*. Values with different letters are significantly ($p \leq 0.05$) different. The bars represent the standard error.

Mucilage films can be used in product packaging. Therefore, the light transmission rate and the transparency of the films are important for their applications. The visible region from 200 to 800 nm was selected to measure the light barrier properties [28]. The light transmission rate (Figure 3) indicates that the film of the cultivar Copena F1 had the highest percentage of transparency. All three films had excellent UV light barrier properties at 200 nm, but as the nanometer value increased, this barrier descended. The film made from the cultivar Jalpa conserved this barrier from 300 to 800 nm. This behavior is similar to that reported by Fang et al. [30], Nouraddini et al. [2] and Thakur et al. [48], due to the nature of the polymer used. Regardless of the plasticizer used, with increasing wavelength, the light transmission rate increased, and it was too difficult to determine the relationship between the transparency and compounds in the films [28]. However, the most transparent film was that using Copena F1 (3.81), and it can be assumed that the light barrier properties of the mucilage decide the transparency of the films. The transparency values of some commonly used synthetic films were reported by Shiku et al. [29]: low-density polyethylene (3.05), orientated polypropylene (1.67), and polyvinyldichloride (4.58). Other reports found similar values for transparency, such as Caamal-Herrera et al. [49] reporting values from 1.53 to 6.38 for films of corn starch-cassava-phaseolus lunatus, and Salinas-Salazar et al. [50] reporting values from 1.20 to 5.44 for films of cactus mucilage-beeswax-grenetine. Therefore, the values obtained suggest that coatings based on the mucilage of *Opuntia ficus-indica* can be used for foods susceptible to light exposure with a better visual representation of the product.

According to the viscosity measurement results, the formulation from the cultivar Copena F1 had the highest values compared to the other two cultivars (Table 2). This is possibly due to the content of monosaccharides in the mucilage of this cultivar. Sáenz et al. [51] noted that the mucilage contains approximately 35–40% arabinose, 20–25% galactose and xylose, and 7–8% rhamnose and galacturonic acid. In addition, water-soluble polysaccharides are long-chain polymers that dissolve or disperse in water, conferring a viscous effect, and are commonly used in foods as thickeners because they increase their viscosity when hydrated [52].

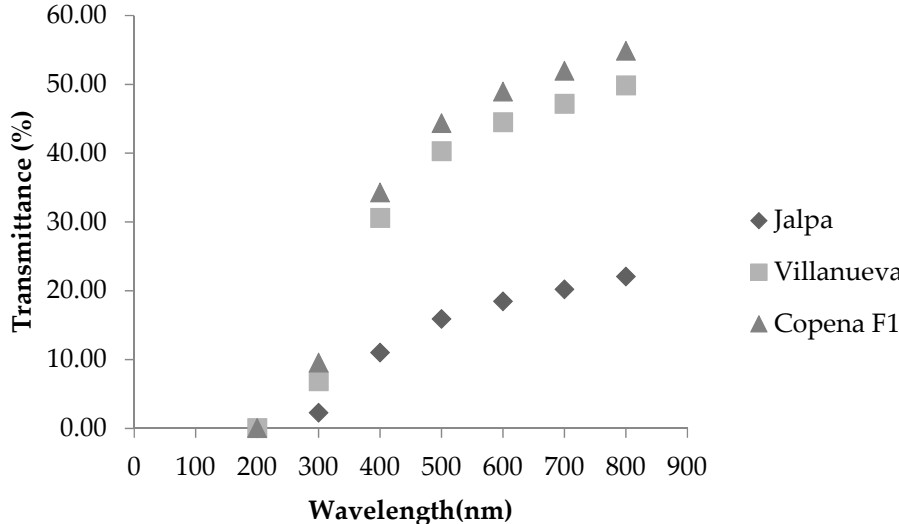

**Figure 3.** Light transmission rate (%) of edible films based on the mucilage of three cultivars of *Opuntia ficus-indica* at wavelengths from 200 to 800 nm.

**Table 2.** Viscosity (mPa × s) of formulations based on the mucilage of three cultivars of *Opuntia ficus-indica*; revolutions per minute (RPM) from 20 to 60.

| RPM | 20 | 30 | 50 | 60 |
|---|---|---|---|---|
| *Jalpa | 30.00 ± 0.00[c] | 29.33 ± 0.12[c] | 27.20 ± 0.0[c] | 26.67 ± 0.23[c] |
| *Villanueva | 72.50 ± 0.46[b] | 65.47 ± 0.12[b] | 56.00 ± 0.10[b] | 52.43 ± 0.15[b] |
| *Copena F1 | 306.33 ± 12.50[a] | 263.00 ± 1.00[a] | 194.00 ± 1.83[a] | 172.33 ± 2.52[a] |

*Cultivar; data are presented as mean ± SD ($n = 5$). Different letters indicate significant differences ($p \leq 0.05$).

Our study shows that each cultivar has different characteristics and demonstrates how they can positively or negatively influence the physicochemical properties of edible films. Each of these properties gives the resulting film different possibilities for use.

## 4. Conclusions

In the current study, edible films containing mucilage from Nopal were successfully fabricated. The cultivar Copena F1 is the best source of mucilage for the formulation of edible films since the resulting films presented low WVP ($1.63 \times 10^{-11}$ g·m$^{-1}$·s$^{-1}$·Pa$^{-1}$), which can prevent the exchange of moisture and reduce the deterioration of the products; high solubility (81.26%); color; yellowness index; transparency; and light transmission rate allowing their use for foods susceptible to light exposure with better visual effect; viscosity allowing their easy handling during formulation and application in several products; and high resistance (4.44 N·mm$^{-1}$), showing that the edible films are capable of maintaining the integrity of the packed product. For this reason, Copena F1 is a potential source for the formulation of edible films based on organic mucilage for application in foods in which visibility of the packaging material is not desired. In addition, experiments will be continued with the cultivar Copena F1 as the best source of mucilage to produce edible films for application to products, and the shelf life of the coated products will be examined.

**Author Contributions:** D.C.G.S., conducted the entire experiment from the collected vegetal material up to the formulation and characterization of the edible films (investigation, resources and writing—original draft preparation). B.L.S., assisted in the development of the whole research part (methodology). G.C.G.M.-Á., contributed as a writer, reviewer, document editor and research project advisor (writing—reviewing and editing). H.R.F., codirected the research project and advised on the agronomic part (supervision). V.H.A.A., participated in the statistical analysis and interpretation of the data (software). R.R. was responsible for the project and directed the research project, as well as reviewing and editing the final version (funding acquisition and project administration).

**Funding:** This research was funded by the Sectorial fund for research, development and forest technological innovation CONAFOR-CONACYT project B-S-65769 "Estandarización del proceso de extraccion de aceites esenciales de especiaes aromaticas: diseño y constriccion de equipo microindistrial".

**Acknowledgments:** The first author thanks CONACYT-Mexico for the grant awarded to develop their Ph.D. in agricultural sciences at the School of Agronomy, UANL.

**Conflicts of Interest:** The authors declare no conflict of interest.

## Abbreviations

| | |
|---|---|
| WVP | Water vapor permeability |
| TGA | Thermogravimetric analysis |
| DSC | Differential scanning calorimetry |
| SEM | Scanning electron microscope |
| FTIR | Infrared spectrometry |
| SD | Stability dispersion |
| PS | Phase separation |
| $\Delta E$ | Total color difference |
| YI | Yellowness index |
| N | Newton |
| UANL | Universidad Autónoma de Nuevo León |
| ASTM | American standard testing materials |
| WVTR | Water vapor transmission rate |

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
