# Peer review of "Formulation and Characterization of Edible Films Based on Organic Mucilage from Mexican Opuntia ficus-indica"

_coatings, doi:10.3390/coatings9080506_

Round 1

Reviewer 1 Report

The article entitled “Formulation and characterization of edible films based on organic mucilage from Mexican Opuntia  ficus-indica" has been carefully reviewed. This is a good attempt towards the use of mucilage from Mexican Opuntia  ficus-indica to formulate an edible film or coating. My only suggestion to authors is to continue the experiment with the cultivar Copena F1 as the best source of mucilage to produce films or coatings in application context and test the shelf life of coated products. I recommend to add this sentence as possibility in "Conclusions" of the paper. 

Reviewer 2 Report

I suggest proofreading by a native speaker.  

1. According to equation 4 the unit for the transparency has to be 1/mm or %/mm or another unit for the thickness. In addition equation 4 irritates me, since a low value for the transparency leads to a better transparency. Thus I suggest leaving this equation and the results for the transparency and discuss the transparency together with figure 3

2. Puncture resistance: In chapter 2 you describe the puncture resistance in the unit N in chapter 3 (results) you present values in N/mm. Do you refer the measured values to the thickness? – Pleas explain the differences in chapter 3

3. The unit for viscosity in Table 2 should be mPa×s

4. Chapter Conclusions is too short, please discuss your results in more detail:

Reviewer 3 Report

The whole paper is very unclear and it should be more specific by highlighting the novelty of the study. The paper is unclearly written and recommended to reject the manuscript. My opinion is based on the 2 bullet points mentioned below:

1.  Authors analyzed only part of the main properties (basic) which are important to conclude as it was done in the conclusions section. It is an interesting topic which can be further developed, however in my opinion described tests, methods and observation are not sufficient.

In the introduction part Authors mentioned about importance of diverse properties evaluation: The characterization consists in evaluate diverse properties of the film like solubility, thickness, viscosity of film forming solution, water vapor permeability (WVP), thermogravimetric analysis (TGA), differential scanning calorimetry (DSC), mechanical properties, SEM analysis, FTIR analysis, color, and others [1, 17, 24, 25]. The importance of this characterization is to understand how the formulation parameters (pH, concentration, additives, varieties, sources, and others) affect the properties of film from barrier, mechanical, physical, thermal to structural properties and others and demonstrate its high potential for use as biodegradable packaging materials for the food industry

TGA, FTIR, SEM, DSC and mechanical properties described in this section should be included in the research part to give the right overview of the formulation and characterization of edible films.

2.  In fact, I am not sure whether I understood the whole content of the manuscript as it is not correctly written in English and it needs extensive editing of English language and style (additionally scientific language should be used, example: page 4 lines 165 – 168).

-  Many repetitions (example: page 3, lines 113-114; lines: 117-119, page 5 lines 180-181 etc)

-  Language mistakes (example: page 2, lines: 46, 46-47, 49, 72, 77 etc)

Moreover, Materials and methods as well as Results and Discussion are not sufficiently and clearly described. The research is not designed appropriately with detailed and clear explanation of the results and discussion part in most of the manuscript. Aim of the research is not clear and there are no tests done related to the quality and shelf life as it was stated in the aim of the research.

Round 2

Reviewer 2 Report

The amendments of the authors are accepted.

Reviewer 3 Report

Major comments were included in the new version of the manuscript as per Reviewer's recommendation.